# A Multi-Disciplinary Approach to Diagnosis and Treatment of Radionecrosis in Malignant Gliomas and Cerebral Metastases

**DOI:** 10.3390/cancers14246264

**Published:** 2022-12-19

**Authors:** Julian Mangesius, Stephanie Mangesius, Matthias Demetz, Christian Uprimny, Gianpaolo Di Santo, Malik Galijasevic, Danijela Minasch, Elke R. Gizewski, Ute Ganswindt, Irene Virgolini, Claudius Thomé, Christian F. Freyschlag, Johannes Kerschbaumer

**Affiliations:** 1Department of Radiation Oncology, Medical University of Innsbruck, 6020 Innsbruck, Austria; 2Department of Neuroradiology, Medical University of Innsbruck, 6020 Innsbruck, Austria; 3Department of Neurosurgery, Medical University of Innsbruck, 6020 Innsbruck, Austria; 4Department of Nuclear Medicine, Medical University of Innsbruck, 6020 Innsbruck, Austria

**Keywords:** radionecrosis, radiotherapy, glioma, brain metastases, MRI, multi-disciplinary, tumor recurrence

## Abstract

**Simple Summary:**

Radionecrosis is a common and rising problem in neuro-oncology. Image interpretation and management of these patients has to be conducted in an interdisciplinary setting in order to offer the best medical care to patients with gliomas or brain metastases. In this article, we provide a state-of-the-art institutional guideline for the current morphological, functional, metabolic and evolving imaging tools to distinguish radionecrosis from tumor recurrence. We also discuss the therapeutic possibilities and give an outlook on future developments to tackle this challenging topic.

**Abstract:**

Radiation necrosis represents a potentially devastating complication after radiation therapy in brain tumors. The establishment of the diagnosis and especially the differentiation from progression and pseudoprogression with its therapeutic implications requires interdisciplinary consent and monitoring. Herein, we want to provide an overview of the diagnostic modalities, therapeutic possibilities and an outlook on future developments to tackle this challenging topic. The aim of this report is to provide an overview of the current morphological, functional, metabolic and evolving imaging tools described in the literature in order to (I) identify the best criteria to distinguish radionecrosis from tumor recurrence after the radio-oncological treatment of malignant gliomas and cerebral metastases, (II) analyze the therapeutic possibilities and (III) give an outlook on future developments to tackle this challenging topic. Additionally, we provide the experience of a tertiary tumor center with this important issue in neuro-oncology and provide an institutional pathway dealing with this problem.

## 1. Introduction

Radiation necrosis (RN) is an uncommon but potentially severe and debilitating late sequela of radiation treatment (RT) of targets inside or near the CNS caused by significant co-irradiation of brain tissue. It is a frequent concern in the treatment of brain metastases, gliomas and AVMs, but is also relevant when targeting meningiomas or vestibular schwannomas. RN provides a challenge from a diagnostic and therapeutic perspective due to the difficulty of differentiating between RN and true tumor progression with conventional MRI and the potentially severe implications in case of misdiagnosis. RN usually occurs between 6 and 12 months after radiation therapy but can appear up to several years later in rare cases. As stereotactic RT is increasingly used for brain metastases and re-irradiation of recurrent gliomas gains popularity in combination with improved survival times, the incidence of RN is also highly likely to increase in the future. While the majority of cases of RN remain asymptomatic (CTCAE grade 1) and are apparent in follow-up MRI only, a significant proportion of cases can manifest with clinical symptoms (CTCAE grade ≥2). These vary depending on the lesion’s location, and include headaches, fatigue, nausea, paresis, sensory deficits, speech deficits and seizures. Incidence rates vary between reports and treatment concepts. 

A frequent indication for the application of intracranial radiation therapy are brain metastases of solid tumors. Local treatment options include single or multisession stereotactic radiosurgery (SRS), whole brain radiation therapy (WBRT) and surgery. For patients with a limited number of brain metastases SRS is increasingly utilized. SRS is now also considered a viable and safe treatment option for up to and sometimes exceeding 10 brain metastases [1]. The increasing application of SRS has also put radiation necrosis into the spotlight [2], since the risk increases with the number of irradiated metastases and the volume of co-irradiated brain tissue. RN represents the main dose-limiting complication in SRS. For this reason the maximum tolerated dose in the RTOG 90-05 protocol was adjusted depending on lesion size (24 Gy for tumors ≤20 mm diameter, 18 Gy for 21–30 mm and 15 Gy for 31–40 mm lesions) [3]. For SRS of brain metastases, Minniti et al. [4] reported a rate of RN of any grade of 24% (14% asymptomatic and 10% symptomatic). Other studies reported lower incidence rates between 11 and 20% [3,5,6,7].

The aim of this article is to provide an overview of multidisciplinary diagnostic and therapeutic methods, including the current morphological, functional, metabolic and evolving imaging tools in order to (I) identify the best criteria to distinguish radionecrosis from tumor recurrence after the radio-oncological treatment of malignant gliomas and cerebral metastases, (II) analyze the therapeutic possibilities and (III) give an outlook on future developments. Furthermore, we present the experience of a tertiary tumor center with this important differential diagnostic neuro-oncological issue and provide an institutional pathway, which is further contextualized and discussed with existing literature.

## 2. Materials and Methods

An institutional pathway is presented that identifies the relevant diagnostic and therapeutic steps a patient undergoes when radionecrosis or tumor recurrence is suspected. These stepwise procedures are then discussed with the existing literature by describing (I) the best criteria for distinguishing radionecrosis lesions from tumor recurrence after radiation oncology treatment of malignant gliomas and cerebral metastases, (II) the therapeutic options and (III) future developments to address this challenging topic. Therefore, a non-exhaustive literature search was performed, and a selection of literature supporting the diagnostic and therapeutic approach is presented in a review.

The institutional practice of a tertiary tumor center is represented in a flowchart (Figure 1) that includes our standard clinical approach, which is then applied individually to our patients. Following this workflow, we discuss cases in the weekly interdisciplinary clinical tumor board. Annually, about 100 cases of malignant gliomas (one third of them as initial diagnoses, the others in cases of recurrence, progress or in cases of radionecrosis) and 130 cases of brain metastasis are treated annually according to this algorithm.

## 3. Current Evidence

### 3.1. Treatment Response

After brain irradiation, residual tumor and tumor recurrence may occur. Additionally, two different types of adverse therapeutic effects can occur: pseudoprogression and radiation necrosis. Differential diagnosis based on imaging between radionecrosis and pseudoprogression is very difficult in early stages, yet both adverse events require completely different therapeutic options.

Radionecrosis is defined as the occurrence of necrotic brain tissue after RT, diagnosed by clinical presentation, MRI, FET-PET/CT and biopsy. Pseudoprogression is defined as a temporary increase in volume diagnosed by clinical presentation, MRI, FET-PET/CT and biopsy without evidence of vital tumors.

Pseudoprogression is thought to be a combination of the therapeutic effect and a collapse of the blood–brain barrier. It occurs between two and five months after the initiation of radiation in approximately 20% of patients with concomitant chemoradiation and follows a self-limiting course. Histologically, neither tumor cells nor inflammatory processes are found [8]. As differentiation from recurrent tumors is difficult, a close monitoring with frequent MRI is recommended so as not to misinterpret a real tumor progression as pseudoprogression [9].

Radiation necrosis usually occurs between three months and one year after radiotherapy [8] and affects about 20% of radiotherapy in GBM patients [10]. The likelihood of its occurrence depends on the radiation dose [11] and the risk is higher after additional chemotherapy [12]. Additionally, MGMT-methylated GBMs were found to be more likely to develop pseudoprogression [13]. Histologically, these lesions have chronic inflammatory reactions and hypoxic necrosis and rarely contain actively proliferating tumor cells [8]. In addition, a rarefication of blood vessels could be demonstrated [14].

Each of the three above-mentioned post-therapeutic conditions has different therapeutic consequences: a tumor recurrence requires, if possible, another operation or at least a treatment modification. Radiation necrosis can be treated conservatively with steroids or, if necessary due to a space-occupying aspect, surgically. A pseudoprogression is usually treated conservatively [8].

To ensure the diagnosis, histopathological proof of active tumor cells is required. Thus, the diagnosis of relapse or pseudoprogression could be delayed. Early detection and the correct interpretation of adverse treatment effects are therefore crucial for patient survival.

Apart from clinical signs and symptoms and routine MRI protocols, several techniques have been developed to aid differentiation, which include perfusion- and diffusion-weighted MRI, FDG- and FET-PET and MRI contrast-clearing analysis. In recent years, there have been several advances in MRI sequences or more specific radionuclides of PET, which have helped to more reliably differentiate the entities described (Figure 2).

### 3.2. Predictive Factors for Radionecrosis

The main predictive risk factors known to play a significant role in the induction of RN are irradiated volume and radiation dose. In radiosurgery planning, the volume of brain receiving a radiation dose exceeding 12Gy (V12gy) or 10Gy (V10Gy) are established measures to estimate the risk of RN [4,6,15]. Korytko et al. [15] reported that the risk of RN correlates with V12, rising from 23% for a volume of <5 cc to 57% for volumes exceeding 15 cc. Similarly, Minniti et al. demonstrated a risk of 47% for V12Gy >10.9 cc. 

Other factors linked to an increased risk of RN are previous WBRT and re-irradiation, male sex, lesion location, tumor biology, as well as systemic and immunotherapies [7,15,16,17]. In a retrospective cohort study including 1939 patients, Miller et al. [17] identified Her2 amplification, BRAF V600+ mutation, ALK rearrangement and lung adenocarcinoma as primary factors associated with RN. Another important consideration in determining the risk of RN is RT fractionation. Fractionated stereotactic radiation therapy, commonly applied in two to five fractions, can lower the risk of RN while providing comparable local control rates [18,19,20,21]. It is therefore increasingly preferred to single session SRS for larger metastases.

Additionally, the use of chemotherapy has been implicated in RN induction, with some reports on the combined chemoradiation in the treatment of gliomas as well as on the use of capecitabine within 1 month of SRS [7,11]. The role of immune checkpoint inhibitors (ICI) in the development of RN is currently inconclusive, with some findings indicating an increased risk with any or some ICI agents [16,22,23], while others demonstrated no difference [24]. Fang et al. [25] found that the type of immunotherapy or timing relative to SRS were not predictive of RN risk in a cohort of 137 melanoma patients treated by ICI with ipilimumab or pembrolizumab with a total of 1094 lesions. However, the incidence of RN was higher in patients receiving chemotherapy within 6 months of RT. Coaco et al. [22] reported an increased risk of RN for patients treated with ICI alone in a cohort of 180 patients receiving SRS of brain metastases, whereas any type of chemotherapy was associated with a lower risk of RN. Estimating the impact of ICI on RN risk is further complicated by the fact that pseudoprogression observed under ICI treatment is challenging to discern from RN or tumor progression without pathological confirmation. There was a higher risk of RN found by Kim et al. [26] in patients treated with targeted therapies, specifically VEGFR tyrosine kinase inhibitors, anti-HER2 as well as EGFR tyrosine kinase inhibitors, whereas chemotherapy did not significantly influence RN development.

### 3.3. Evaluation of Treatment Response by MRI

Due to the expected high rate of tumor progression or treatment effects after radiotherapy, monitoring for radionecrosis or tumor recurrence is crucial. The determination of radiological outcome requires observation periods of between 4 and 9 months [4,27]. Close-meshed monitoring by repeated MRI is therefore routinely performed: from the start of diagnostic imaging (for tumor characterization and treatment planning), after treatment (as a baseline examination for comparisons with follow-up MRI) and in repeated follow-up MRI examinations performed every 3 months or when medically indicated (to assess tumor control and adverse treatment effects).

## 4. Institutional Practice

### 4.1. Magnetic Resonance Imaging Protocol

Routine clinical MR examinations in our institution are conducted in standardized time intervals, usually every 3 months or upon new neurological symptoms. MR images are routinely acquired using a 1.5 or 3 Tesla MRI with a standard multichannel head coil and include the sequences described in Table 1. Additionally, for the routinely performed MRI protocols for brain tumors, 3D T1-Magnetization-Prepared Rapid Gradient-Echo (MPRAGE) with a contrast agent 80 min after the initial contrast agent application can be performed to calculate contrast clearance and accumulation, as explained herein.

#### 4.1.1. Structural MRI—Glioma

Unfortunately, in structural MRI, including T2-weighted images, native and post-contrast T1-weighted images, progression, pseudoprogression and radiation necrosis may have a similar presentation: collapse of the blood–brain barrier with marginal contrast and hyperintensive perifocal edema in T2-weighted sequences [8]. A tumor progression could be diagnosed on a case-by-case basis with MRI if the contrast agent uptake can be detected within the first 12 weeks after the completion of chemoradiation beyond the 80% isodose line. Less specific signs of progression are an increase in contrast agent accumulation of more than 25% at 12 weeks despite the use of corticosteroids or an increase in T2/FLAIR hyperintensity with antiangiogenic therapy [28]. 

The classical appearance of radionecrosis is a mass which enhances with a central necrotic area. Contrast enhancement reflects a disturbance of the blood–brain barrier secondary to the endothelial irradiation-induced damage. As the lesions tend to occur at the site of the maximum irradiation dose, they are generally expected in the immediate vicinity of the tumor site and around the excision cavity. Affected sites are predominantly in the white matter. White matter is particularly vulnerable to the secondary ischemic consequences of post-irradiation vasculopathy, resulting in perivascular coagulative necrosis, as the deep white matter has a relatively poor blood supply from the cortical arteries and is located in the watershed area of the arterial blood supply that is susceptible to ischemic damage. Consequently, the white matter arcuate fibers which also receive cortical arterial supply are more resistant to radionecrosis and are therefore usually affected later in the disease process [29].

The morphological appearances of contrast enhancement described for gliomas after radiotherapy, including irregular, annular or nodular uptake are non-specific and contradictory [30,31].

Contrast uptake in radionecrosis may be nodular, linear or curvilinear [11]. Heterogeneous and annular enhancement has been given catchy names such as “cut-green pepper”, “soap bubbles” or “Gruyere” cheese appearances [32,33]. One important decision criterion is the interpretation of lesion boundaries: Blurred plumed boundaries are suggestive of radionecrosis compared to the nodular boundaries with clear edges suggestive of tumor recurrence [34]. If the cortex is affected, contrast enhancement can be gyriform [32]. The change in contrast enhancement varies depending on the dynamic of the underlying pathophysiological process in radionecrosis lesions.

Importantly, while these effects are predominantly expected in the initial target site, single or even multifocal post-irradiation contrast enhancement can also be present distant to the initial tumor. This has been described several centimeters away, in the corpus callosum, the contralateral hemisphere, the subependymal regions and the posterior fossa [32], which makes image interpretation especially challenging in cases of malignant infiltrative gliomas. 

Hypointensities suggestive of hemorrhagic changes are often observed on T2-weighted echo gradient images. These were reported in 53% of cases of radionecrosis using the conventional T2* sequence [35] and in 80% of cases of radionecrosis using magnetic susceptibility imaging [36].

The relevance of the space-occupying effect is typically negligible considering the size of the lesion. Yet, accompanying reactive vasogenic edema can be extensive and cause a significant space-occupying effect. In cases of possible radionecrosis, some lesions progressively enlarge with a transient increase of cytotoxic edema over a few months. However, some lesions remain stable before regressing and others regress from the beginning [37]. Therefore, not only tumor progression, but also symptomatic active expansile lesions due to radionecrosis may require early surgery to reduce the space-occupying effect and provide a definite diagnosis.

Overall, in many cases, the diagnosis remains unclear with structural MRI. The morphological features of radionecrosis lesions are very similar to those of a recurrent tumor, including the type of contrast enhancement, site (both in the tumor site and remote) and possible transient deterioration [32]. 

#### 4.1.2. Structural MRI—Metastases

MRI changes are observed on MRI from 6 weeks after radiotherapy up to 15 months [38]. A transient increase in volume of over 20% of the first post-treatment review is seen in 30% of cases after stereotactic RT for cerebral metastases [38] and is largely dependent on the radiosensitivity of the primary tumor and the immune reaction associated with the inflammation and necrosis: in radiosensitive tumors (lung, breasts and colon) metastases experience a transient increase in volume, while metastatic lesions of non-radiosensitive tumors remain relatively stable. Usually, these increases in volume are asymptomatic and only require monitoring. As strong immune responses are associated with increased survival and control of cancers, this transient increase in size can predict favorable prognoses [39].

While this transient increase in volume post-radiation is modest, an increase of more than 65% compared to the pre-therapeutic volume suggests a recurrence or continued tumor activity (sensitivity: 100%, specificity 80%) [31]. 

A qualitative method to distinguish both appearances is the correlation between the boundaries of the lesion seen on enhanced T1-weighted and T2-weighted imaging [40]. Correlations indicate tumor recurrence and non-correlations (“T1/T2 mismatch”) suggest radionecrosis. The sensitivity of the “T1/T2 mismatch” was reported to be 83% with a specificity of 91%.

Additionally, the amount of edema present on T2-weighted imaging by calculating the ratio between T2-weighted hyperintensity volume/T1-weighed volume after enhancement, can also be considered. When over 10, this approach reaches a positive predictive value of 92% for radionecrosis [41]. 

#### 4.1.3. Diffusion-Weighted MRI

Diffusion-weighted MRI (DWI) can represent the movement of free water [42] and thus indirectly make statements about cellularity. Tumor recurrence has a higher cellularity with pleiomorphic nuclei and a denser network of cytoplasmic processes than radiation necrosis, which is paucicellular with increased water in the interstitial spaces due to fibrotic lesions and inflammatory effects involving macrophage and polynuclear cell influx [27]. Significantly lower ADC was suggested in tumor recurrence compared to radionecrosis, accordingly [27,43]. 

The method is nevertheless not very well suited for differentiation, since surrounding edema leads to impairments [44]. Moreover, hemorrhagic changes resulting in hemosiderin deposition may also reduce signal by a T2* or T2 dark-through effect [45]. Additionally, ADC values may, however, also be increased by micro-angiogenesis or necrotic effects. The proposed apparent diffusion coefficient values (ADC) to distinguish recurrence from radionecrosis have been contradictory [27,43,46,47], while others found no significant difference between radionecrosis and tumor recurrence [48]. 

The ADC ratio (rADC or mean ADC of the contrast enhancing area/mean ADC of the same area in the contralateral hemisphere) has been reported to be more discriminatory than absolute ADC values with a significantly higher rADC in radionecrosis (rADC = 1.82) compared to tumor recurrence (rADC = 1.43) [43]. Yet, assessment of mean and maximum ADC values revealed lower values in tumor recurrences, while no significant difference was found except for the maximum ADC (1.68 × 10^−3^ mm^2^/s mean compared to 2.3 × 10^−3^ mm^2^/s for radionecrosis) [27]. In contrast, Sundgren et al. reported significantly higher mean ADC values in the recurrence group (1.27 × 10^−3^ mm^2^/s) than in the radionecrosis group of patients (1.12 × 10^−3^ mm^2^/s) [46,49]. 

One solution might be to consider ADC values in the T2-weighted hyperintense area without contrast enhancement, referred to as perilesional edema. The mean ADC in radionecrosis is not significantly different between the enhancing area and the neighboring area, while in recurrences of an infiltrative tumor the ADC is higher outside of the contrast enhancement [43,44]. 

#### 4.1.4. Perfusion-Weighted Imaging—Glioma

With perfusion-weighted MRI (PWI), local tissue perfusion can be measured and compared with healthy areas. By measuring dynamic contrast agent accumulation, the relative cerebral blood volume (rCBV) can be determined. In tumor relapses, increased rCBV can be found due to increased metabolic activity and neoangiogenesis [50]. Radiation necrosis results in occlusive vasculopathy leading to ischemia and decreased rCBV. However, rapidly growing tumors can lack an adequate blood supply, leading to necrosis and also decreased rCBV [51]. Antiangiogenic substances can cause a pseudoprogress that has a decreased rCBV [52]. 

#### 4.1.5. Perfusion-Weighted Imaging—Metastases

The tumor capillaries of cerebral metastases resemble the ultrastructure of the original tumor more than the cerebral tissue capillaries. Increased tortuosity, lack of maturity, increased permeability and a lack of blood–brain barrier are indicators for alternate image interpretation using macromolecular contrast agents based on relative cerebral blood volume (rCBV), the relative amplitude of the peak (rHP) and the percentage signal recovery (PSR) [31,53,54]. Accordingly, a significant increase in average and maximum rCBV is observed in metastatic recurrence, with cutoff values ranging from 1.52 to 2.1 [31,53]. Still, rCBV values of <1.35 have also been observed in irradiation lesions [54]. Additionally, DCE-MRI has been shown to differentiate pseudoprogression from progression in growing lesions in patients with melanoma brain metastases who received immunotherapy [55].

This large overlap of CBV values renders image interpretation challenging. This effect may be explained by tumor heterogeneity, the similarity between microvascular density from metastatic recurrences and hyperplastic dilated blood vessels in post-radiation changes, the co-existence of radionecrosis lesions with tumor recurrence (50% radionecrotic tissue has been found in tissue samples from metastatic recurrence), magnetic susceptibility artifacts due to petechial hemorrhage caused by the irradiation (artificially reducing the rCBV of a recurrence) and melanin as an artifact impeding the analysis of the CBV [37].

#### 4.1.6. (1H) Magnetic Resonance Spectroscopy

Hydrogen 1 (1H) magnetic resonance (MR) spectroscopy (MRS) enables non-invasive in vivo quantification of metabolite concentrations in the brain. The structural brain tissue degradation induced by radiotherapy is caused by early alterations in metabolic activity (Figure 3). These changes precede the development of neurocognitive symptoms and cannot be visualized on structural images in MRI during the early stages. 

Metabolites affected by ongoing changes include the neuronal marker N-acetylaspartate (NAA), the concentrations of which decrease as a consequence of cell death by apoptosis or neuronal dysfunction. Secondly, alterations in the biosynthesis of cell membranes and metabolic turnover are reflected by an increase in choline (Cho). Contrarily, the marker of energy metabolism Creatinine (Cr), is supposed to be unaffected by radiation damage in some literature [53]. Consequently, a small rise in the Cho peak and Cho/Cr ratio is observed in brain tissue developing radionecrosis [47,56,57,58,59,60,61,62]. 

While in cases of tumor recurrence a fall in NAA can also be observed, MRS depicts a dominant rise in the Cho peak due to proliferation. The most widely calculated ratios in clinical routines for gliomas are Cho/Cr and Cho/NAA [47,57]. Additionally, to distinguish metastatic recurrence from radionecrosis, the calculation of the choline ratio (rCho = Cho lesion/Cho contralateral healthy area) has been suggested, whereby an increased ratio correlates with tumor recurrence and with higher diagnostic accuracy compared to the conventional ratios (Cho/Cr and Cho/NAA).

Spectroscopic analysis is a promising method; however, it bears several limitations, such as lesions located close to bones due to magnetic susceptibility artifacts [60]. Additionally, spectroscopy can reliably distinguish between tissues experiencing pure radionecrosis or a pure recurrence. The co-existence of both, however, poses a major challenge to interpreting the resulting spectra [58] as the metabolite values are averaged within the studied voxel. This is especially challenging in cases of monovoxel studies. Multivoxel spectroscopy may potentially offer a more complex, yet more thorough examination of the affected areas. Tumor recurrence with typical tumor spectrum profiles may furtherly be more accurately depicted by means of spectroscopy in areas which do not enhance, as well as in the adjacent white matter [59].

#### 4.1.7. When the Difficult Becomes Even more Challenging: Bevacizumab and Systemic Therapies

Imaging evaluations of treatment responses are especially challenging after the administration of Bevacizumab and similar systemic therapies, as there might be an influence of common systemic treatments on the accuracy and readability of contrast clearance and perfusion MR images. Bevacizumab (trade name Avastin), is an inhibitor of the vascular endothelial growth factor (VEGF), aimed at inhibiting neoangiogenesis. VEGF is produced by the tumor and not only promotes neoangiogenesis but also reduces the effectiveness of gap junctions and creates fenestrations in the endothelium of existing brain capillaries, leading to edema and enhancement. The amount of VEGF produced has been shown to correlate with tumor grade. 

Unlike the USA and Japan, Bevacizumab is not approved for first- or second-line treatment of GBM in the European Union. Nevertheless, it is most frequently used as a second line treatment in Europe, due to a lack of other agents and good clinical response in terms of symptomatic relief. In our neuro-oncological practice, Bevacizumab is used after tumor board approval for second-line treatment of patients with recurrent GBM after re-resection and re-irradiation (as appropriate). Bevacizumab is known to change imaging patterns and contrast medium uptake dynamics. It is likely to introduce a so-called pseudoresponse, which refers to the phenomenon of tumors appearing to respond to a specific treatment on imaging criteria, even though the lesion actually remains stable or has even progressed.

In brain tumors imaging follow-up, especially in high grade gliomas such as glioblastoma, a rapid decrease in contrast enhancement and edema can be observed in a short period of time after the administration of antiangiogenic agents (e.g., Bevacizumab and Cediranib), often without any significative change in actual tumor size (as visualized by a T2 non-enhancing tumor) or diffusion/perfusion studies [63,64,65].

The changes seen in pseudoresponse, which can be observed as a rapid reduction in enhancement and vasogenic edema in imaging, are largely mediated by changes in blood–brain barrier permeability rather than antiangiogenic effects, meaning that relying on enhancement and T2 signal change can be misleading when interpreting follow up MRI studies. 

MR spectroscopy, MR perfusion (particularly cerebral blood volume) and diffusion-weighted imaging (DWI) are particularly important in assessing the presence of residual, but now non-enhancing, tumors. 

Despite the fact that these advanced MRI sequences have helped in differentiating pseudoresponses from a true response, imaging follow-up is still required [63]. Essential to correct interpretation is the availability of multiple previous imaging studies and information relating to the type and timing of therapy. 

The influence of targeted or immune checkpoint therapies on image interpretation has not been investigated yet. However, to date there is no known impact of those substances on GBM. The results of imaging studies in targeted or immune-checkpoint modulating treatments may allow for future guidelines on how to interpret imaging results.

### 4.2. Evolving MRI Techniques

Recent developments have provided radiologists with advanced MR techniques, which are currently being extensively investigated and have recently been shown to be useful in the imaging of neuro-oncology patients. Although still investigational techniques, they have been shown to have potential routine clinical application and should therefore be incorporated in decision making in the future.

#### 4.2.1. Delayed Contrast Extravasation MRI and Quantitative Imaging

Despite all of their limitations, all of these above-mentioned sequences are part of standard MRI protocols.

Brain metastases and radiation effects after SRS have been shown to bear a characteristic and statistically significantly different signal intensity (SI) time course, as assessed by manual regions of interests (ROI) drawing on sequential gadolinium enhancement MRI including MR studies at 2 (TP1), 15 (TP2) and 55 (TP3) min after administering the contrast agent [66]. 

To perform delayed contrast extravasation MRI, an additional short MRI scan is added >1 h after a standard contrast enhanced MRI scan. For the analysis subtraction, maps are obtained in which T1-MR images acquired 5 min postcontrast are subtracted from T1-MR images acquired 80 min postcontrast. T1-MRI of the second time point are therefore registered to the location of the first time point. Finally, subtraction maps are calculated by voxel-by-voxel subtraction of the early images from the late images. The derived maps depict the spatial distribution of the contrast accumulation/clearance. Exemplarily, in the case of healthy blood vessels, as an effect of the contrast clearance from the blood, the signal decreases with time, and hence the subtraction maps between these two images show negative values (arbitrarily color-coded as blue in the maps). In contrast, in case of contrast accumulation, the maps show positive values (red) [67]. This is based on the concept that active tumor tissue is characterized by the effective clearance of the contrast agent, whereas in necrotic tissue the contrast agent accumulates over time, which can be visually observed and manually measured. This methodology thus promises better differentiation of treatment-induced effects versus real tumor progression. A diagnostic sensitivity of 100% and a positive predictive value of 92% have been reported, thus demonstrating the feasibility of delayed contrast extravasation MRI in brain tumors for the differentiation of necrosis from vital tumor tissue [67]. This recent study could show that delayed contrast extravasation MRI has a higher sensitivity regarding necrosis and PPV regarding brain tumor activity compared to DCS-perfusion MRI, as was shown in patients with histopathological confirmation [67]. Recent developments have allowed for the automatization of image fusion and image subtraction, enabling high resolution analysis of contrast agent clearance versus accumulation, which can then be color-mapped in treatment response assessment maps (TRAMs). The delayed contrast extravasation MRI does not require contrast agent application in addition to the routinely applied amount of contrast agents for routine clinical work up to determine contrast enhancement. 

In addition to delayed contrast extravasation MRI, quantitative sequences and arterial spin labelling (ASL) sequences are tools for the quantification of image data in radiology and thus pave the way for a more objective and reproducible non-invasive diagnosis [40,68,69,70]. ASL provides an absolute quantification of cerebral blood flow (but not CBV) and is not affected by capillary leakage, which leads to an underestimation of CBV and flow in perfusion sequences after gadolinium enhancement. ASL was reported to have a high diagnostic sensitivity to distinguish radionecrosis from recurrence [71]. 

Upcoming technologies might influence future approaches on MR imaging protocols for patients affected with primary brain tumors. DSC perfusion is used in standard brain tumor MR protocols at most tumor centers. Due to the added invasiveness of contrast application, associated socio-economic costs and increasing awareness of potential Gadolinium accumulation, DSC perfusion might potentially be replaced by arterial spin labelling MR perfusion, which might have potential as a diagnostic aid in the differential diagnosis of tumor recurrence, radionecrosis and pseudoprogression. 

Furthermore, T1-, T2- and T2*/QSM-mapping and arterial spin labelling sequences are a tool for the quantification of image data in radiology and thus pave the way for a more objective and reproducible non-invasive diagnosis [40,68,69,70,72]. Importantly, T1- and T2-mapping and arterial spin labeling sequences do not require the application of contrast agents. However, their benefit in differentiation of post therapeutic effects has not been investigated yet.

#### 4.2.2. (31P) Magnetic Resonance Spectroscopy

By means of phosphorous-based MRS (31P-MRS), various phosphorus-containing metabolites can be measured in vivo: energy-related metabolites (the energy metabolites inorganic phosphate (Pi), phosphocreatine (PCr) and adenosine triphosphate (ATP)) and cell membrane-related phospholipids (the mobile membrane phospholipid precursors phosphomonoesters (PME) and their breakdown products and intracellular signaling molecules phosphodiesters (PDE), which are related to membrane turnover) [71].

Differences in energy and membrane metabolism have been detected with 31P-MRS between contrast-enhancing (CE) tumors and the contralateral hemisphere, normal-appearing areas of the brain, as well as brain tissue from healthy controls and further during therapy [73,74,75,76,77,78,79,80,81,82].

Consequently, 31P-MRS can potentially be applied for differentiation between radionecrosis and tumor recurrence, yet, this warrants further investigations. However, the application of 31P-MRS is complicated by the fact that 31P-metabolism varies throughout the brain, with age and between sexes, which poses important practical implications for the application and interpretation of 31P-MRS especially in such complex metabolic changes as are introduced by radiation [83].

#### 4.2.3. Deep Neural Networks

Furthermore, an alternative approach might be the development of deep evolution learning for monitoring and analyzing the treatment effects of radiotherapy. In that context, the performance of clinical routine and advanced MR sequences could be included for diagnostic and prognostic examinations. The term deep evolution learning is used for finding data-driven models based on deep neural networks (NNs) to understand and analyze the evolution of images. Multi-channel data collected via different measurements, which are obtained with routine and advanced sequences in MRI, could potentially extract key figures to provide a learned model for the dynamic evolution of the disease under the external influence of radiotherapy. By enabling a more precise and objective diagnosis, deep evolution models offer the opportunity to individualize treatment planning and to guide therapy. The identification of sensitive parameters and key features that drive the temporal evolution limits the risks and could potentially serve as a warning system to assist the physician. The methodological aspects of such a biomarker, in order to be implemented as a potential outcome measure in research and clinical routine, i.e., sequence selection, needs to be investigated. By implementing a local image analyses pipeline, with growing well-defined data gained in a clinic every day, we expect tertiary centers to thereby not only achieve a robust and easily feasible diagnostic and prognostic marker with a therapeutic value but also enforce a well-structured team, including medical doctors, computer scientist and statisticians, in order to efficiently perform image analysis in both clinical routines and scientific advancement. This approach may even spare patients from the invasiveness of brain biopsy.

### 4.3. Nuclear Medicine—PET

#### 4.3.1. PET Tracers

While FET is used in our clinic as a main tracer in neuro-oncology, other tracers exist. 2-Fluor-2-desoxy-D-glucose (FDG) represents the most common agent to display increased metabolism; unfortunately, its role within the brain is limited due to the high baseline metabolism. Amino acid-based tracers have the advantage of a superior differential between tumors and normal brains [84].

L-methyl-^11^C-methionine (^11^C-MET), O-2-^18^F-fluoroethyl-L-tyrosine (^18^F-FET), and 3,4-dihydroxy-6-^18^F-fluoro-L-phenylalanine (^18^F-FDOPA) are common amino acid-based tracers that have been reported in glioma [85,86,87,88]. Alternative tracers such as FLT or Cholin are used in other centers with similar diagnostic yields. Unfortunately, the comparison of different radiotracers results in controversial conclusions with no clear benefit of one tracer over the others [89]. Altogether, amino acid PET tracers are helpful in planning biopsies of inhomogeneous tumors and the monitoring of treatment responses in glioma as well as the distinction of radionecrosis and tumor recurrence. The best evidence exists for ^18^F-FET PET and ^11^C-MET PET as it is used in most centers. ^11^C-MET requires an on-site cyclotron because of the short half-life of only 20 min, whereas ^18^F-FET has a half-life of about 110min and can be delivered to distant centers. 

Therefore, ^18^F-FET PET is the tracer we use in our institution.

#### 4.3.2. ^18^F-FET—PET

[^18^F] fluoroethyl-l-tyrosine (^18^F FET)-PET can be used to detect an existing GBM with great reliability. An important limitation in the application of the method in the course of therapy is the passive influx of the radiopharmaceutical due to a blood–brain barrier disorder in the lesion [90]. In cases of radiographic or clinical suspicion of local recurrence or radiation necrosis, brain biopsy is the procedure of choice, but also ^18^F-FET-PET can be employed for diagnosis [28,91,92,93]. 

For PET-imaging, O-2-^18^F-Fluoroethyl-L-Tyrosine (^18^F-FET) is used. ^18^F-FET, a radioactive Fluorine labelled tyrosine analog, is transported into cells by amino acid transporters, which are overexpressed on the tumor cell surface. Fluorine (^18^F) decays with a half-life of 110 min, enabling scintigraphic imaging. ^18^F-FET is indicated in patients with known or suspected glioma for the characterization of tumors, the monitoring of treatment and the detection of viable tumors after treatment. In accordance with guidelines, patients are injected with an activity ranging from 180 to 250 MBq via an intravenous line. The effective dose is 16µSv/MBq, resulting in an effective dose of 2.9–4.0 mSv for the administered activity. The first image is acquired 10 min after the tracer injection, followed by images at time points 20 min, 30 min, 40 min and 50 min post-injection for dynamic PET analysis. For attenuation correction a low-dose CT (100 kV, 10–80 mA, slice thickness of 3.75 mm) of the brain is acquired. Images are reconstructed iteratively with an ordered subset maximization algorithm (OSEM). For image analysis a visual interpretation is performed: a high tracer uptake in a nodular lesion being suggestive of tumors; a low tracer uptake in a rim-like pattern being suggestive of post-treatment changes. In addition, a semi-quantitative evaluation of the lesions is applied by measuring the intensity of the tracer uptake of normal brain tissue and a lesion-to-background ratio. Time-activity curves are also generated using SUVmax values of the lesions at the predefined five different time points of image acquisition. The time to peak is defined and time-activity curves will be assigned to one of three curve patterns: (1) constantly increasing tracer uptake, (2) tracer uptake peaking between 20 and 40 min post-injection followed by a plateau and (3) tracer uptake with an early peak (<20 min post-injection) followed by a constant decrease. Curve pattern 2 and 3 are suggestive of tumors, whereas curve pattern 1 is more likely to be present in inflammatory lesions such as post-therapeutic changes.

An example of the multimodal diagnosis of radiation and tumor progression is shown in Figure 4.

## 5. Treatment Strategies

### 5.1. Medical Treatment

A mild to moderate increase in size after radiation therapy may be observed in about 30% of all patients between a few weeks and up to 15 months after intervention [38]. As many of these lesions tend to stabilize or even regress over time, close observation is a feasible option in asymptomatic patients [94]. Radiation-induced damage to the vascular endothelium resulting in a breakdown of the blood–brain barrier with inflammation has been targeted with steroids as a short-term therapy [95,96], leading to stabilization in many symptomatic patients. However, a clear dosing scheme is not consistent throughout the literature and long-term use of steroids is not feasible due to the numerous adverse effects (e.g., hyperglycemia, osteoporosis and psychiatric disturbances). Treatment with anticoagulants and antiplatelet agents to counteract the impaired microcirculation and hypoxia in radionecrosis has also failed to provide a satisfactory response [97]. Hyperbaric oxygen therapy has also been investigated in the treatment and prevention of RN by promoting perfusion and angiogenesis, but the evidence is limited to small case series and the specialized facilities needed and the significant time commitment do not allow for broad use [98,99,100,101].

The role of hypoxia-induced factor 1 α (HIF-1 α) in the initial step of the development of radiation necrosis results in an overproduction of VEGF. Subsequent cerebral edema is caused by fragile angiogenesis. Bevacizumab as a potent anti-VEGF monoclonal antibody has been introduced in 2007 by Gonzales et al. with satisfactory responses [102] and showed effectiveness in further studies, irrespective of tumor type and radiation modality [103,104,105]. Nowadays, it is the standard therapy in the treatment of radiation necrosis, even if cerebral hemorrhage and thromboembolic complications may be observed [106]. However, as outlined above, the antiangiogenic changes introduced by Bevacizumab and similar agents pose a major challenge for image interpretation and correct differential diagnoses.

### 5.2. Surgical Treatment

Surgical treatment is usually necessary to either ensure a histological diagnosis, to address a space-occupying effect or to resect tumor tissue if oncologically indicated. If radiation necrosis seems plausible according to the above-mentioned imaging modalities, surgery is rarely indicated. The differentiation between real tumor progression and pseudoprogression, however, poses a significant challenge and has important therapeutic implications. If the diagnosis cannot be established by radiological means only and/or uncertainty is too high, a histological verification is often required. Surgical removal can be performed in easily accessible lesions and is indicated especially in symptomatic lesions with elevated intracranial pressure [107]. The neurological condition, however, can worsen after these procedures [108], so that, although often not very effective, conservative treatment should be considered primarily. Therefore, stereotactic biopsy for histological proof represents an important tool to confirm the diagnosis with high accuracy prior to systemic therapy, particularly in surgically inaccessible lesions [109]. Typical histopathological presentation of necrotic cells with gliosis and aggregation of lymphocytic borders with thrombosed vessels can proof radionecrosis (Figure 5).

## 6. Conclusions

Given the data and understanding of the underlying histological processes of tumor recurrence, pseudoprogression and radiation necrosis, in which changes in vascular density and consequent contrast media washout appear to play an important role, routinely applicable imaging modalities as well as new technologies appear promising in differentiating the three entities. Thus, it could be possible to optimize the management of patients with brain tumors and metastases by means of greater diagnostic reliability and to speed up necessary treatment modifications, to improve surveillance and to detect changes of brain lesions with MRI more rapidly, to promptly offer efficacious treatment strategies. It may be possible to reduce the necessity for invasive diagnostic procedures and reduce the rate of misdiagnosis of treatment effects after radiation therapy. More efficacious therapies and consequently longer average survival times have led to higher rates of the possible treatment responses described. This trend will most likely continue in the future. 

In anticipation of new MR techniques potentially limiting the necessity of invasive procedures such as biopsies, a close multidisciplinary interaction and cooperation between the teams of Neurology, Neuroradiology, Neurosurgery, Nuclear medicine and Radiooncology is crucial. Readings of MRI and PET demonstrating signs of radionecrosis or tumor progress should be carefully reviewed by all involved physicians in the context of the overall clinical scenario. A multidisciplinary approach is vital not only during tumor board sessions but also in daily clinical practice in order to offer the best medical care to patients with gliomas or brain metastases suffering from radiotherapeutic side effects.

## Figures and Tables

**Figure 1 cancers-14-06264-f001:**
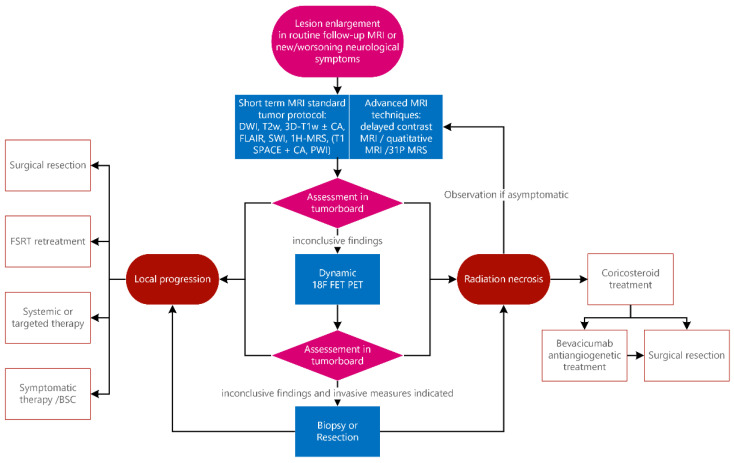
Diagnostic and therapeutic algorithm for suspected tumor recurrence or radiation necrosis. In cases of tumor enlargement on MRI or neurologic symptoms suggestive of tumor progression, a short-term MRI scan is performed using advanced new techniques. If the results are inconclusive, a dynamic ^18^F-FET PET scan or, in selected cases, a biopsy or surgical resection may be performed. All results are discussed in an interdisciplinary tumor board (with specialists in neurology, neurosurgery, neuroradiology, nuclear medicine and radiation oncology), where a final decision is made regarding the differential diagnosis of radiation necrosis and/or progression and the assignment of further diagnostic and therapeutic steps. Radiation necrosis not responsive to corticosteroids can be treated with bevacizumab or surgical resection in rare cases. Abbreviations: diffusion-weighted images (DWI), T2-turbo spin echo (T2-TSE; T_2_w), fluid-attenuated inversion recovery (FLAIR), 3D T1-Magnetization-Prepared Rapid Gradient-Echo (MPRAGE; 3D-T_1_w), without and after (−/+), contrast agent (CA), T1-sampling perfection with application-optimized contrast using different flip angle evolutions (T1-SPACE; in case of metastases), susceptibility-weighted imaging (SWI), perfusion-weighted imaging (PWI; in case of glioblastoma) and Hydrogen 1 magnetic resonance spectroscopy (1H-MRS).

**Figure 2 cancers-14-06264-f002:**
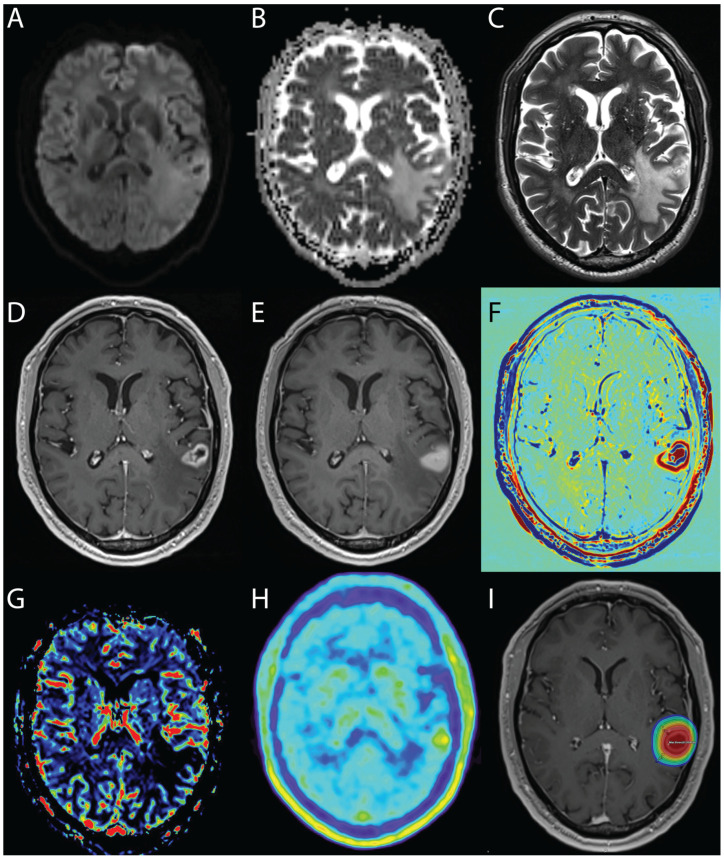
Example of radionecrosis in the parietal lobe of the left hemisphere after radiosurgery of a metastasis from non-small-cell lung cancer with 20 Gy on the 80% PTV marginal isodose: no diffusion restriction on diffusion-weighted images, the mean ADC in radionecrosis is not significantly different between the enhancing area and the neighboring area (**A**,**B**), non-correlation between the boundaries of the lesion seen on enhanced T1-weighted and T2-weighted imaging (“T1/T2 mismatch”) (**C**), non-specific morphological appearances of contrast enhancement in post-contrast T1-weighted images (**D**), which increases diffusely after 80 min in the delayed contrast extravasation MRI (**E**), as the contrast agent accumulates over time in necrotic tissue, color-coded in red in the treatment response assessment map (TRAM) (**F**), decreased rCBV as a result of occlusive vasculopathy leading to ischemia (**G**) and low tracer uptake in in the ^18^F-FET-PET examination (**H**). The lesion is located within the high dose area of the radiation therapy (**I**).

**Figure 3 cancers-14-06264-f003:**
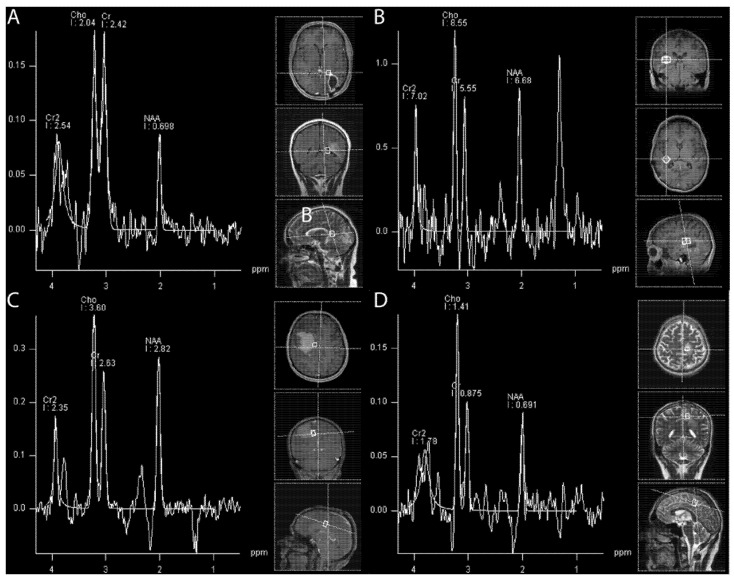
Assessment of treatment effect in (1H) Magnetic Resonance Spectroscopy. (**A**): True progression in a case of glioblastoma: reduced NAA, increased Cho. Cho/NAA ratio over cut-off, with a mean of 2.72. (**B**): Radionecrosis after radiosurgical treatment of a metastasis: Cho/NAA ratio with a mean of 1.46. (**C**): Pseudoprogression (glioblastoma): Cho/NAA ratio under 1.47–2.11 and Cho/Cr ration under 0.82–2.25. (**D**): Mixed image between pseudoprogression and true progression in a glioblastoma case.

**Figure 4 cancers-14-06264-f004:**
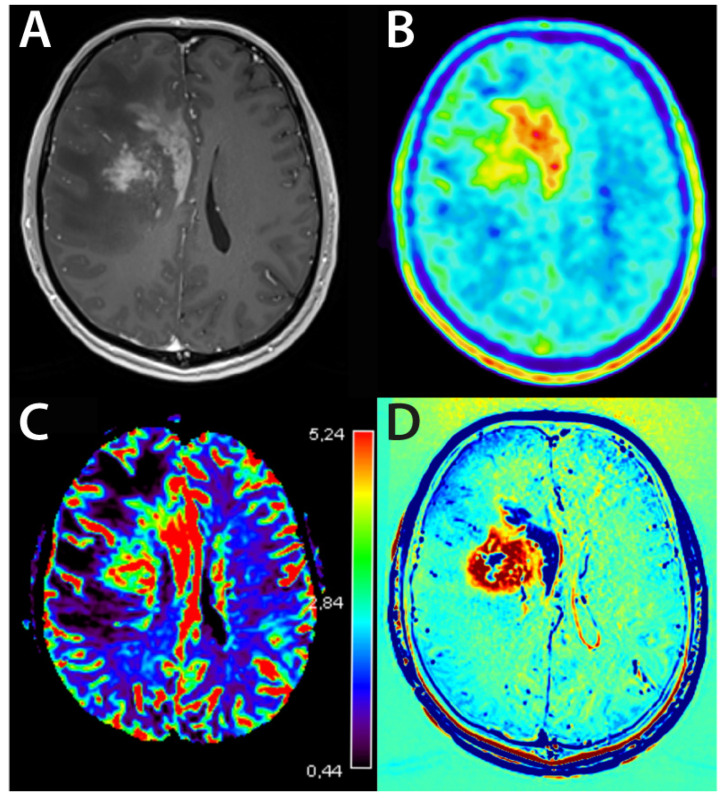
Example of both radiation necrosis and tumor progression in a case of glioblastoma after radiochemotherapy. (**A**): structural MRI, (**B**): FET-PET, (**C**): perfusion-weighted MRI, (**D**): delayed contrast MRI.

**Figure 5 cancers-14-06264-f005:**
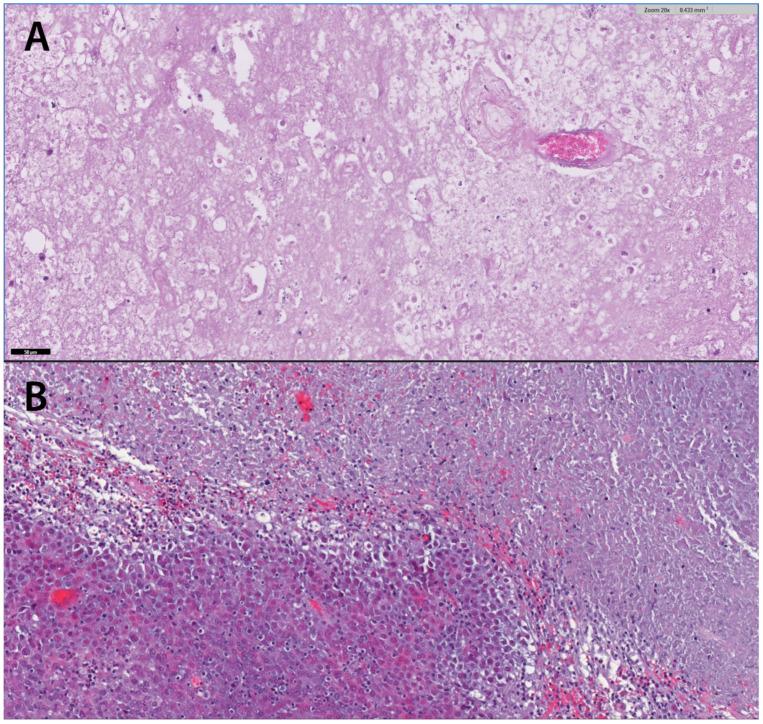
(**A**): Typical appearance of an RN in a patient treated due to progressive CE mass after resection and focal radiotherapy for a melanoma BM. Necrotic areas and shadows of vessels without tumor cells. HE staining 20×. (**B**): Necrotic tissue adjacent to vital tumor cells with eosinophilic cytoplasm in a patient with NSCLC. Tumor cell nuclei are variable in size. HE staining.

**Table 1 cancers-14-06264-t001:** Protocol of MRI Sequences.

Sequence	TA (mins)	Voxel Size (mm³)	TR (ms)	TE (ms)	FA	FOV (mm²)	AM	ST (mm)
**Native**								
Transverse diffusion-weighted sequence (DTI) †	5:28	2.0 × 2.0 × 2.0	9600	92	90°	250 × 250	128 × 100%	2
Native transverse T1-weighted magnetization-prepared rapid gradient-echo sequence (T1-MPRAGE)	3:37	0.9 × 0.9 × 1.0	2210	3	8°	220 × 179	256 × 100%	1
Coronal T2-weighted fluid-attenuated inversion recovery (FLAIR)	3:14	0.7 × 0.7 × 3.0	8000	87	150°	220 × 172	320 × 70%	3
Transverse gradient-echo susceptibility-weighted imaging sequence (SWI)	3:04	0.9 × 0.9 × 1.8	27	20	15°	220 × 172	256 × 90%	1.8
Transverse Hydrogen 1 magnetic resonance spectroscopy (1H-MRS) ††	6:53	10.0 × 10.0 × 15.0	1700	135	90	160 × 160	n.a.	15
**Gadolinium contrast agent application**								
Transverse dynamic T2*-weighted susceptibility contrast (DSC) perfusion (in case of glioblastoma) †††	1:42	0.9 × 0.9 × 4.0	1600	30	90°	220 × 220	128 × 100%	4
Transverse T2-weighted turbo-spin echo sequence (T2 TSE)	3:13	0.6 × 0.6 × 2.0	5800	95	150°	220 × 179	384 × 70%	2
Post-contrast transverse T1-weighted magnetization-prepared rapid gradient-echo sequence (MPRAGE)	3:37	0.9 × 0.9 × 1.0	2210	3	8°	220 × 179	256 × 100%	1
Post-contrast sagittal T1 SPACE fat sat dark blood sequences (in case of metastases)	5:55	1.0 × 1.0 × 1.6	600	9.5	variable	256 × 256	256 × 90%	1
**80 min after contrast agent application**								
Post-contrast transverse T1-weighted magnetization-prepared rapid gradient-echo sequence (MPRAGE) ††††	3:37	0.9 × 0.9 × 1.0	2210	3	8°	220 × 179	256 × 100%	1

† b-factors: 0/1000 s/mm^2^; 30 diffusion directions; †† Standard mode is given; MR sequence needs to be individually adapted to the pathology and anatomy of each patient; ††† A standard single dose (3 mL/s, 0.1 mL/kg) of gadobutrol 604 mg/mL (1.0 mmol/mL) is injected intravenously using an automatic injection system 9 s after starting the DSC-MRI sequence. No preload contrast dose is administered. †††† Can be performed supplementarily to evaluate delayed contrast clearance versus accumulation; Abbreviations: TA: time of acquisition; TR: time to repetition; TE: time to echo; FA: flip angle; FOV: field of view; AM: acquisition matrix; ST: slice thickness and n.a.: not applicable.

## Data Availability

Not applicable.

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
