# Peer review of "A Multi-Disciplinary Approach to Diagnosis and Treatment of Radionecrosis in Malignant Gliomas and Cerebral Metastases"

_cancers, 2022, doi:10.3390/cancers14246264_

Round 1
Reviewer 1 Report
The authors present a well-written and comprehensive review of the imaging modalities (MRI and PET) for recognizing radiation necrosis. I think that the manuscript could be helpful for physicians involved in brain tumor management. I have some minor issues related to the role of different, potentially useful amino acid: MET PET still play a role in different centers, as well as FET PET. Do the authors think that alternative tracers such as FLT (or, despite the very limited applications up to date, also Cho) could play in the future a similar role? (see i.e. what happens for brain tumor: Guglielmo P et al. [18F] Fluorothymidine Positron Emission Tomography Imaging in Primary Brain Tumours: A Systematic Review. Curr Med Imaging. 2022;18(4):363-371)
Author Response
Response to Reviewer 1 Comments
Point 1: The authors present a well-written and comprehensive review of the imaging modalities (MRI and PET) for recognizing radiation necrosis. I think that the manuscript could be helpful for physicians involved in brain tumor management. I have some minor issues related to the role of different, potentially useful amino acid: MET PET still play a role in different centers, as well as FET PET. Do the authors think that alternative tracers such as FLT (or, despite the very limited applications up to date, also Cho) could play in the future a similar role? (see i.e. what happens for brain tumor: Guglielmo P et al. [18F] Fluorothymidine Positron Emission Tomography Imaging in Primary Brain Tumours: A Systematic Review. Curr Med Imaging. 2022;18(4):363-371)
Response 1: Thank you for your comment. The major players in Neurooncology are L-[methyl]-11C-methionine (MET), 18F-fluoroethyl-tyrosine (FET), 18F-fluoro-L-dihydroxy-phenylalanine (FDOPA) and 11C-alpha-methyl-L-tryptophan (AMT). Altogether, aminoacid PET tracers are helpful in planning biopsies of inhomogeneous tumors and monitoring of treatment response in glioma as well as distinction of radionecrosis and tumor recurrence. The best evidence exists for FET PET and CMET PET as it is used in most centers. We have revised the manuscript to reflect the evidence for alternative tracers (see line 553-562 of the revised manuscript).
Reviewer 2 Report
This is a well-written and well-structured review about a common and important issue in neuro-oncology dealing with the diagnosis and treatment of radionecrosis and pseudoprogression in brain tumors. The authors provide . The authors are presentig their institutional approach for the multidisciplinary pathway to solve this issue.
I have a few points that should to be addressed:
Major points:
1. Please provide further informations on different MRI parameters (e.g. time to repetion, time to echo, slice thickness, flip angle, field of view, temporal resolution and the parameter of the dynamic bolus used)
2. I am a little confused about the technique of delayed extravasation MRI. Is this identical to a dynamic contrast enhanced MRI (DCE-MRI) which is performed in addition to a dynamic susceptibility contrast MRI (DSC-MRI)?. Please clarify. If so, please include recent literature on DCE-MRI, e.g. Umemura et al. Journal of Neuro-Oncology (2020) 146:339–346, among others.
3. The manuscript could be further improved by adding more illustrative figures for MR.-spectroscopy or dynamic FET-PET images showing the differences in the appearance of a radionecrosis vs a pseudoprogression.
4. The same holds true for immunhistochemistry. A picture of a radionecrosis is shown, but an illustrative picture of a pseduoprogression is missing.
Minor points:
1. Please specify the title: radionecrosis of what?
2. Please remove on page 3: 3. Results line 104 - 107
3. Figure legend 2 needs to be re-formated.
Author Response
Response to Reviewer 2 Comments
This is a well-written and well-structured review about a common and important issue in neuro-oncology dealing with the diagnosis and treatment of radionecrosis and pseudoprogression in brain tumors. The authors are presentig their institutional approach for the multidisciplinary pathway to solve this issue.
I have a few points that should to be addressed:
Major points:
Point 1: Please provide further informations on different MRI parameters (e.g. time to repetion, time to echo, slice thickness, flip angle, field of view, temporal resolution and the parameter of the dynamic bolus used).
Response 1: We thank the reviewer for the valuable comment. We have expanded table 1 in the revised manuscript and included specific details for all MR sequences used, including a detailed description of the bolus.
Point 2: I am a little confused about the technique of delayed extravasation MRI. Is this identical to a dynamic contrast enhanced MRI (DCE-MRI) which is performed in addition to a dynamic susceptibility contrast MRI (DSC-MRI)?. Please clarify. If so, please include recent literature on DCE-MRI, e.g. Umemura et al. Journal of Neuro-Oncology (2020) 146:339–346, among others.
Response 2: We have included a more comprehensive description of the methodology used for the delayed contrast MRI (see line 460-469 of the revised manuscript).
Point 3: The manuscript could be further improved by adding more illustrative figures for MR.-spectroscopy or dynamic FET-PET images showing the differences in the appearance of a radionecrosis vs a pseudoprogression.
Response 3: We thank the reviewer for the suggestion. We have included a new figure (figure 4) to show the different presentation of radiation necrosis and tumor progression in PET, perfusion MRI and delayed contrast MRI. Additionally we have added another figure (figure 3) to show different MR-spectroscopic characteristics for tumor progression, radiation necrosis, pseudoprogression, and mixed progression and necrosis.
Point 4: The same holds true for immunhistochemistry. A picture of a radionecrosis is shown, but an illustrative picture of a pseduoprogression is missing.
Response 4: We thank the reviewer for the valuable suggestion. We have added a histopathologic image of a real tumour progression to better contrast the presentation of radiation necrosis vs. progression. Pseudoprogression is rarely an indication for biopsy unless further developing into a progressive radiation necrosis.
Minor points:
Point 5: Please specify the title: radionecrosis of what?
Response 5: We have changed the title of the revised manuscript to provide a more accurate representation of the reviews scope. “A multi-disciplinary approach to diagnosis and treatment of radionecrosis in malignant gliomas and cerebral metastases”.
Point 6: Please remove on page 3: 3. Results line 104 - 107
Response 6: Thank you for pointing this out, we have corrected the mistake.
Point 7: Figure legend 2 needs to be re-formated.
Response 7: The legend has been reformated accordingly.
Round 2
Reviewer 2 Report
No further comments. All my issues raised have been thouroughly answered.